# The longitudinal relation between adolescents' learning outcomes and internalizing symptoms: The role of ADHD symptoms

**Linda Visser**[1,2,3], **Jan-Henning Ehm**[3,4], **Janin Brandenburg**[5]*

**1** Department of Developmental Psychopathology, Behavioural Science Institute, Radboud University, Nijmegen, The Netherlands, **2** DIPF | Leibniz Institute for Research and Information in Education, Frankfurt am Main, Germany, **3** Center for Research on Individual Development and Adaptive Education of Children at Risk (IDeA), Frankfurt am Main, Germany, **4** Institute for Psychology, Heidelberg University of Education, Heidelberg, Germany, **5** Department of Rehabilitation Sciences, TU Dortmund University, Germany

* janin.brandenburg@tu-dortmund.de

## Abstract

### Objective

The literature on bivariate relations between learning outcomes and symptoms of anxiety, depression, and ADHD is extensive. Much less research has been done into the trivariate longitudinal relations between learning outcomes, internalizing symptoms, and ADHD symptoms, which was the focus of the current study.

### Method

The sample from the Adolescent Brain Cognitive Development (ABCD-) Study was largely representative for the US population in terms of, i.e., race and ethnicity and included 11,867 children aged 9 or 10 years (47.8% female; 52.2% male) at the start.

### Results

The results of bi- and trivariate latent change score modelling with four timepoints showed that school records and symptoms of anxiety or depression were related, but school records did not predict these symptoms at the next timepoint, nor the other way around. ADHD was associated with both school records and symptoms of anxiety/ depression. Depression symptoms were a negative leading indicator of subsequent changes in ADHD.

### Conclusions

The results suggest that ADHD symptoms do not form the main explanatory factor for the relation between school records and internalizing symptoms. They imply that it is important to recognize signs of depression at an early stage, so that such secondary problems can be prevented. Future research is needed to find underlying risk factors

**Data availability statement:** The data underlying the results presented in the study stem from the ABCD study and are available from the NIH Brain Development Cohorts (NBDC) Data Sharing Platform (https://www.nbdc-datahub.org/abcd-study). The preregistration for this research are available at https://osf.io/yw5jv/overview?view_only=c258b240352a4d-4fa3b309a79d656ce9. The analytic code necessary to reproduce the analyses is available at https://osf.io/gw8d3/overview?view_only=5bff385bc7074df8b9bd0bb75bf1e5f2.

**Funding:** The author(s) received no specific funding for this work.

**Competing interests:** The authors have declared that no competing interests exist.

**Abbreviations:** ADHD, attention deficit hyperactivity disorder.

that can explain comorbidity, which can help identifying children with increased risk at an early stage.

---

## Introduction

If children have persistent difficulties in reading, spelling, and/or mathematics despite appropriate schooling and in absence of other disorders that could explain these difficulties, the diagnosis of a specific learning disorder (SLD) [1] applies. In the literature about SLD, the presence of comorbidities with different types of behavioral problems is well established. The estimates on prevalence rates for these comorbidities vary widely depending on the criteria used. In general, symptoms of attention deficit/hyperactivity disorder (ADHD) seem to be the most common comorbidity for SLD [e.g., 2,3].

SLD is related to higher rates of internalizing behavior problems as well, including symptoms of both anxiety and depression [4,5]. Especially in cases in which an SLD is present in multiple learning domains, the risk for multiple forms of behavioral problems is high [5]. This yields the question if the comorbidity between SLD and internalizing problems could be explained by co-occurring symptoms of ADHD, which indeed seems to be the case for both anxiety and depression symptoms, at least partly [6]. However, the risk for developing internalizing problems still appears higher in cases of comorbid SLD and ADHD, compared to ADHD alone [7].

In the context of the multiple deficit model, neurodevelopmental disorders are believed to arise due to the probabilistic contribution of various risk factors. Comorbidity between different disorders can be explained by common risk factors [8]. To increase our understanding of neurodevelopmental disorders and comorbidity based on the multiple deficit model, longitudinal studies are needed to evaluate the relation between various symptoms and risk factors over time.

Therefore, the current study focused on the longitudinal relation between learning outcomes, internalizing symptoms, and ADHD symptoms in a general sample of early adolescents. Increased knowledge about this relation in the general sample can help increase our understanding of comorbidity on the clinical level. Indeed, ADHD- and internalising symptoms appear to relate not only to SLD on a clinical level, but also to learning outcomes in general [9,10]. In addition, not only clinical, but also subclinical symptoms have been associated with reduced quality of life in children [11]. Therefore, we took both symptom levels into account by focusing on the continuum of symptoms and school outcomes, rather than on neurodevelopmental disorders as categories. We focused on early adolescence, as this developmental period is known to be a critical period for internalizing problems to arise [12].

### Longitudinal trajectories of learning outcomes and internalizing symptoms

Before studying the longitudinal relation, an understanding is needed about the longitudinal trajectories of the individual constructs. With respect to learning outcomes, longitudinal studies in the United States have shown slight declines, at least from

grade 6 onwards [13–15], which might be related to the transition to middle school and was found irrespective of gender [15]. With respect to anxiety, earlier studies have shown a decrease in symptoms during early adolescence [16], followed by a subsequent increase in symptoms [e.g., 17,18], showing a discontinuous trajectory. The decrease in symptoms seems to be higher in children with higher initial symptom levels [16,18]. Important to keep in mind is that the individual variability in the development of anxiety symptoms is high [19].

With respect to the development of depression symptoms in early adolescence, inconsistent results have been found. McLaughlin and King [16] found no significant change in adolescents between 10 and 15 years of age. Cohen and colleagues [17] followed children from 7 to 15 years of age and found a slight decrease in symptoms until the age of 12, followed by a slight increase. Ormel and colleagues [18] found an increase from age 15 onwards. Based on a systematic review [20], depression symptoms seem to peek around 15–17 years of age, thus decreasing again afterwards. The trend in symptom change seems to be independent from initial symptom levels [16,21]. For both anxiety and depression, McLaughlin and King [16] found higher symptom levels in girls, but no gender differences with respect to change. It seems that symptoms of negative affect, anhedonia, and anxious arousal develop relatively independently from each other, at least in late adolescence [22].

How does the development of anxiety and depression symptoms relate to school outcomes? Levels of anxiety and achievement are negatively correlated [19], but this relation has been found to be weak ($r = -.06$) [23]. Certain types of anxiety might be related to school outcomes more than others. For example, Morin and colleagues [24] found that children with anxiety related to school transition showed the lowest school outcomes. For depression, based on a meta-analysis, the influence of symptoms on subsequent school outcomes is small but significant (pooled $r = -0.19$), independent from gender, also after adjusting for various confounding variables [25].

Results regarding the direction of effects are mixed, with some studies mainly finding an effect from achievement on later depression [26], others mainly finding an effect from depression on subsequent achievement [27], again others finding a negative relation (random intercept), but not a dynamic one [28]. Weidman and colleagues [29] did find a transactional relation, with higher levels of depression and anxiety predicting lower school outcomes as well as the other way around. In a study by Chen et al. [30] among elementary school children, the results of a cross-lagged panel model showed effects of school outcomes on depression symptoms and vice versa, but those of random-intercept cross-lagged panel models only showed an effect of depressive symptoms on school outcomes.

Explanations for the bivariate relation between internalizing symptoms and school outcomes are given by the *academic incompetence hypothesis* (academic incompetence leads to or worsens internalizing problems) and the *adjustment erosion hypothesis* (internalizing symptoms have a negative effect on school outcomes) [e.g., 28]. Based on the current literature, there is slightly more support for the *adjustment erosion hypothesis*, but a firm conclusion is not possible because most studies only focus on one direction of effect or have methodological shortcomings [31].

### The role of ADHD-symptoms

Alternative explanations for the relation between internalizing symptoms and school outcomes are given by the *transactional effects model* and the *shared risk hypothesis* (a third variable increases the risk for problems in both domains) [e.g., 31]. Focusing on this last hypothesis, one variable that has been linked to school outcomes as well as internalizing symptoms are symptoms of ADHD. Symptoms of ADHD [32] as well as an ADHD diagnosis [e.g., 33] appear relatively stable. It is well established that ADHD symptoms influence learning outcomes [e.g., 34–36]. Less clear is if the disadvantage for affected children increases with age [35,36] and if ADHD symptoms are related to the growth in school performance [34].

Earlier studies have also shown that ADHD symptoms have a significant influence on subsequent levels of anxiety [37] as well as on changes in anxiety [38]. Vice versa, anxiety influences subsequent levels of ADHD in adolescents [37], but not in young children [39].

Where the literature on trajectories of the individual constructs and on bivariate relations is extensive, much less research has been done into the trivariate relation between learning outcomes, internalizing symptoms, and ADHD symptoms. As mentioned before, earlier correlational research found that ADHD symptoms partly explain the relation between anxiety and learning outcomes [6]. In addition, a 4-year longitudinal study among college students [40] identified depression symptoms as one of the strongest predictors for college outcomes in students with ADHD, which were lower compared to those of students without ADHD.

## Research questions and hypotheses

Based on the literature described above, the current study is confirmatory in nature and based on the following three sets of preregistered [41] research questions and hypotheses.

First, we evaluated how the shape of the developmental trajectories looks like for:

a. school records. We expected to find a decline from grade 6 onwards, which is related to the transition to middle school [14,15].

b. anxiety symptoms. We expected to find a decrease in anxiety symptoms with age (negative linear slope) [e.g., 16–18] and greater declines in children with higher anxiety levels at baseline (a negative correlation between the intercept and slope) [e.g., 16,18].

c. depressive symptoms. We expected to find no change at first and an increase in symptoms starting around age 12 (grade 7) [17,21]. We expected this change from grade 7 onwards to be caused by a linear slope factor [21], not dependent on the previous time point [17]. Also, we did not expect differences in change depending on the initial level of symptoms [16,42].

In addition, we hypothesized for each of the variables a to c that the rank order stability between adjacent time points (1-year interval) would be high, whereas the stability between more distant time points would be lower.

Second, we evaluated the longitudinal relation between school records and:

a. anxiety symptoms. We expected a negative correlation between the initial levels of anxiety and achievement (significant correlation between the baseline factors) [e.g., 19,29]. In addition, we expected to find low achievement to predict higher levels of anxiety in the next school year [29].

b. depressive symptoms. On a theoretical basis we expected to find effects of depression on subsequent achievement and vice versa (both an academic incompetence effect and an adjustment erosion effect) [28]. In addition, we expected a negative correlation between the initial levels of school records and depression [28].

Third, we evaluated the trivariate dynamic relation between ADHD-symptoms, school records and:

a. anxiety symptoms. We hypothesized that:

 i. ADHD symptoms have a significant influence on both the starting level (intercept) of school records and on changes in school records (increasing disadvantage) [35,36];

 ii. The starting levels of ADHD and anxiety are positively correlated [5]. In addition, ADHD symptoms have a significant influence on subsequent levels of anxiety and anxiety has a significant influence of subsequent levels of ADHD [37]. We also expected an influence of ADHD symptoms on the change in anxiety [38];

 iii. if a relation between school records and subsequent anxiety is found under step 2, this can be explained by ADHD symptoms [6].

b. depression symptoms. We hypothesized that:

 i. ADHD symptoms have a significant influence on the starting level of depression [5] as well as on the change in depression (slope) [43,44];

 ii. if relations between school records and subsequent depression symptoms and between depression and subsequent school records are found, these can be explained by ADHD symptoms [6,40].

## Materials and methods

### Data and sample

We used the data from a large-scale longitudinal study in the United States into the biological and behavioral development of children from 9–10 years of age onwards: the Adolescent Brain Cognitive Development Study [45], funded by the National Institutes of Health (NIH). The baseline measure (T0) took place between September 2016 and August 2018 [46] with a sample of 11,867 children aged 9 or 10 years old. The current study is based on the data from data release 5.0 regarding psychopathology, neurocognition, and school records from the first four time points: T0 to T3. Because data for school records were missing for the fourth time point, we did not include any data from T4. Although data for the other variables were available, using alternative methods such as imputation was not an option, because school records formed an essential element in the analyses and were missing for all children at T4. Most of the children were 12–13 years old at T3. Table 1 shows descriptive statistics for gender, age, and grade at the four time points. We did not exclude any cases.

 The 21 research sites that collected data for the ABCD study obtained ethical permission via the central Institutional Review Board (cIRB) at the University of California, San Diego or via a local IRB [47]. Informed consent was obtained from both parents and children. The sample was recruited with the aim to obtain representativity for the US population with respect to age, gender, race and ethnicity, socio-economic status, and urbanicity. This aim was largely met, although children from rural areas were slightly underrepresented. For more information about the recruitment and sample, we refer to Garavan et al. [46].

### Instruments

Learning outcomes were operationalized as the answer of the parent to the question "Please look at the list and pick the line that best describes your child's grades in school last year". The answer options were given in various ways to specify grades, in numbers, letters, as well as descriptions. For the analysis, we used the numbers: 1 = Excellent (A), 2 = Good (B), 3 = Average (C), 4 = Below Average (D), and 5 = Struggling a lot (F). If parents chose the answer option *ungraded* or *not applicable*, we recoded this answer into a missing value.

**Table 1. Descriptive Statistics of the ABCD Sample at Baseline (T0), T1, T2, and T3.**

| T | N (%) | Age in years;months | | | Gender n (%) | | | Grade n (%) | | | | | | |
|---|---|---|---|---|---|---|---|---|---|---|---|---|---|---|
| | | M | SD | Range | female | male | other | 3 | 4 | 5 | 6 | 7 | 8 | 9 |
| 0 | 11,867 (100%) | 9;11 | 0;8 | 8;11–11;1 | 5,676 (47.8%) | 6,190 (52.2%) | 1 (0.0%) | 2,032 (17.1%) | 5,415 (45.6%) | 3,984 (33.6%) | 358 (3.0%) | | | |
| 1 | 11,219 (94.5%) | 10;11 | 0;8 | 9;8–12;5 | 5,341 (47.6%) | 5,863 (52.2%) | 15 (0.1%) | | 1,958 (17.5%) | 4,963 (44.2%) | 3,796 (33.8%) | 399 (3.6%) | | |
| 2 | 10,972 (92.4%) | 12;0 | 0;8 | 10;7–14;0 | 5,180 (47.2%) | 5,722 (52.2%) | 70 (0.6%) | | | 1,469 (13.4%) | 4,820 (43.9%) | 3,856 (35.1%) | 671 (6.1%) | |
| 3 | 10,335 (87.1%) | 12;11 | 0;8 | 11;5–14;9 | 4,788 (46.3%) | 5,319 (51.5%) | 228 (2.2%) | | | | 2,009 (19.4%) | 4,544 (44.0%) | 3,123 (30.2%) | 345 (3.3%) |

*Note.* T = time point; In the numbers regarding Grade, only those > 100 were included.

Symptoms of depression, anxiety, and ADHD are based on parent's responses on the Child Behavior Checklist (CBCL) [48]. The CBCL is a parent-report questionnaire for assessing the behavior of school-aged children between 6 and 18 years of age. It consists of a total of 112 items that the parent answers on a 3-point Likert scale (not true, somewhat/ sometimes true, very true/ often true). We used the raw score on the DSM-oriented scales Depressive problems, Anxiety problems, and Attention Deficit/Hyperactivity problems, respectively. For these scales, the manual [48] reports a test-retest reliability of $r = .80 - .93$ and Cronbach's alpha of.72 −.84. Higher raw scores indicate higher levels of problems.

With respect to validity, the CBCL has been shown to discriminate well between referred and non-referred children and correlate moderately to strongly with corresponding scales of the Behavior Assessment System for Children (BASC; $r = .52 - .77$) [49] and the Conners Parent Rating Scale Revised (CPRS-R; $r = .71$) [50]. We standardized the scores based on the data at baseline to enhance comparability and interpretation.

## Data analysis

Data were prepared using RStudio [51] and analyzed using Mplus, version 8.6 [52]. In a first step, we calculated descriptive statistics for all variables of interest, taking into account gender. Because the results hinted at gender differences, we included gender in the further analyses as a covariate. Gender was coded as male = 0 and female = 1 in the dataset.

**Research question 1.** We examined the shape of the developmental trajectories with latent change score modelling. Specifically, we tested the following models for school records, anxiety symptoms, and depressive symptoms:

a) No change model (equivalent to the intercept-only model in the latent growth model [LGM] framework)

b) Constant change model (equivalent to the linear growth model in the LGM framework)

c) Linear change model (equivalent to the quadratic growth model in the LGM framework)

d) Dual change score model

We constrained the residual variances of the indicators to be equal across time for model parsimony, but released this constraint for single or all indicators if modification indices indicated a model misspecification concerning this matter. The expectation was that the developmental trajectories would represent a linear latent change model or a dual change model.

To answer the research questions, we examined (a) the mean and the variance of the intercept factors and the slope factors, (b) the covariances between the intercept and the slope factors, and (c) the estimates of the proportional change parameters. We used the final longitudinal models obtained at this stage as a basis for the models related to research question 2.

**Research question 2.** We used two different Bivariate Latent Change Score Models (BLCSM) [53] for testing the longitudinal relation between school records on the one side and depressive and anxiety symptoms, respectively, on the other. Based on the final latent change score models of the previous stage we included coupling (across variable) effects. The models were built stepwise, from simple to complex:

1) No coupling model (i.e., fixing both coupling parameters to 0)

2) Two unidirectional coupling models where either one of the coupling parameters is estimated

3) Full coupling model where both coupling parameters are jointly estimated

We put equality constraints on the coupling parameters suggesting constant dynamics. These constraints could be released for a single or all indicators if modification indices indicated a model misspecification concerning this matter. To evaluate the significance of the coupling parameters, we compared the resulting models 1–3 using the Satorra-Bentler scaled chi-square difference test to examine which of the specified models fit the data best.

For the final models, we also examined the associations between the intercept factors and slope factors across constructs (i.e., how level and change in school records is associated with level and change in anxiety/depression and vice versa). In bivariate change models, the intercorrelation between all slope factors and intercept factors is usually freely estimated per default. Concerning this assumption of correlated growths and intercepts, we, in addition, tested alternative model restrictions of the final bivariate change models, in that we specified (a) the slope covariances at zero, (b) slope intercept and covariances across constructs at zero, and (c) all connections over time at zero. Again, the resulting models were compared with the Satorra-Bentler scaled chi-square difference test.

The final bivariate latent change score models obtained at this stage formed the basis for the models related to research question 3.

**Research question 3.** We added ADHD symptoms to the model longitudinally in the BLCSMs obtained at stage 2, resulting in trivariate LCSMs. We followed the steps as described above for the bivariate model (Models 1–3) for the relation of ADHD with both anxiety/ depression and with school records. To answer the Research Questions 3.a.iii and 3.b.ii (e.g., if ADHD explains a possible relation between anxiety/depression and school records), we examined the following combination of effects:

Combination 1:

a) an effect from ADHD on school records at the next time point,

b) a correlation between ADHD and depression/ anxiety at the same time point, and

c) a nonsignificant or decreased effect from depression/ anxiety to school records at the next time point, compared to the model without ADHD;

Combination 2:

a) an effect from ADHD to depression/ anxiety at the next time point,

b) a correlation between ADHD and school records at the same time point, and

c) a nonsignificant or decreased effect from school records to depression/ anxiety at the next time point, compared to the model without ADHD.

Children are nested in families and families are nested within sites. For study site, the ICCs ranged between 0.00 und 0.02. When we took into account the sites as a covariate, none of the regression coefficients were statistically significant. Therefore, and because of the low ICCs, we decided that controlling for study site was not necessary and used a two-level model controlling for the family level using "TYPE = COMPLEX" in Mplus.

## Statistical criteria

We used an alpha level of .05 and judged model fit using the chi-squared test. Because the chi-squared test is sensitive to large sample sizes, we additionally used the criterion chi/df ratio < 2 [54]. Regarding fit indices, we used the root mean square error of approximation (RMSEA) ≤ .06 and comparative fit index (CFI) ≥ .95. Model comparison was based on the adjusted chi-squared difference test using the Satorra-Bentler scaling correction [55] and the mentioned fit indices. In addition, the dual change score model was only chosen in case of a significant proportional change component.

For all models, we used the robust Maximum Likelihood (MLR) estimator in MPlus. To test the specific hypotheses formulated, we looked if the result of interest (intercept, slope, constant change, proportional change, or coupling effect) was significant.

## Results

### Descriptives

Table 2 shows the descriptive statistics for the school records and levels of depression, anxiety, and ADHD symptoms for each time point by gender. For gender we used the variable indicating the sex of the child, because this was the variable with the least missing data. The three cases with missing values for sex had indicated being *male* as gender and data were recoded accordingly. The sample size differed slightly for each variable due to missing data. See S1 Table for the exact sample sizes.

### Developmental trajectories

Table 3 shows the model fit for the various models evaluated with respect to the developmental trajectories (research question 1).

**School records.** As shown in Table 3, the RMSEA had an optimal value for the constant change model with random slope but was almost identical for the dual change score model. The Satorra-Bentler chi-squared (SB-$\chi^2$) test showed the dual change score model (d1, see Table 3) to be the optimal model. More specifically, it fitted better compared to the no growth model (SB-$\chi^2$ = 576.51; df = 5; $p < .01$) as well as the constant change models with fixed (SB-$\chi^2$ = 521.12; df = 3; $p < .01$) or random slope (SB-$\chi^2$ = 7.89; df = 1; $p < .01$). The linear change model did not fit better than the constant change model with random slope (SB-$\chi^2$ = 4.77; df = 2; $p = .092$), which had a better fit than the one with fixed slope (SB-$\chi^2$ = 593.42; df = 2; $p < .01$).

The model results of the dual change score model (d1) showed an intercept of 1.771 ($p < .01$; variance 0.446, $p < .01$) and a negative constant change component (−0.299, $p = .016$; variance 0.043, $p < .01$). Lower values for school records reflect better achievement, which means that this negative value reflects an improvement in grades over the years. The estimate for the proportional change component, reflecting the change in school records between time points based on

**Table 2. *M* (*SD*) for Standardized Scores per Time Point and by Gender as well as *t*-test Results for Differences Between Females and Males.**

| Time point | Gender | School records | Depression | Anxiety | ADHD symptoms |
|---|---|---|---|---|---|
| 0 | Female | 1.60 (0.74) | −0.06 (0.94) | 0.00 (0.99) | −0.17 (0.89) |
| | Male | 1.77 (0.84) | 0.06 (1.05) | 0.00 (1.01) | 0.16 (1.07) |
| | Diff. | $t(10,930) = 11.67$** | $t(11,855) = 6.33$** | $t(11,810) = 0.19$ | $t(11,760) = 18.21$** |
| | Total | 1.69 (0.80) | 0.00 (1.00) | 0.00 (1.00) | 0.00 (1.00) |
| 1 | Female | 1.58 (0.75) | 0.01 (1.04) | 0.01 (1.00) | −0.21 (0.86) |
| | Male | 1.78 (0.84) | 0.11 (1.13) | −0.01 (1.02) | 0.09 (1.04) |
| | Diff. | $t(10,441) = 12.67$** | $t(11,198) = 5.13$** | $t(11,134) = −1.03$ | $t(11,104) = 16.44$** |
| | Total | 1.68 (0.80) | 0.06 (1.09) | 0.00 (1.01) | −0.06 (0.97) |
| 2 | Female | 1.59 (0.77) | 0.09 (1.12) | −0.07 (0.96) | −0.27 (.82) |
| | Male | 1.78 (0.85) | 0.11 (1.10) | −0.13 (0.94) | 0.03 (1.00) |
| | Diff. | $t(10,264) = 12.14$** | $t(7,975.1) = 1.01$ | $t(7,982.7) = -2.66$** | $t(7,993.1) = 14.79$** |
| | Total | 1.69 (0.82) | 0.10 (1.11) | −0.10 (0.95) | −0.11 (0.93) |
| 3 | Female | 1.59 (0.78) | 0.24 (1.29) | −0.04 (.99) | −0.27 (0.81) |
| | Male | 1.81 (0.88) | 0.17 (1.16) | −0.15 (0.93) | 0.03 (1.00) |
| | Diff. | $t(9,398.4) = 13.31$** | $t(7,356.7) = -2.42$* | $t(7,476.1) = -4.74$** | $t(7,556.1) = 14.32$** |
| | Total | 1.71 (0.84) | 0.20 (1.23) | −0.10 (0.96) | −0.11 (0.93) |

*Note.* Diff. = Difference between females and males; * = *p* <.05; ** = *p* <.01.

**Table 3. Fit Statistics for the Models for Evaluating the Developmental Trajectories.**

| | Model | χ2 (df) | SCF | RMSEA [90% CI] | CFI | TLI | SRMR | AIC | sa-BIC |
|---|---|---|---|---|---|---|---|---|---|
| SR (N = 11,799) | a. no growth | 657.306 (14) | 1.5148 | .062 [.058,.067] | .941 | .958 | .043 | 82,205.294 | 82,222.086 |
| | b1. constant change | 613.080 (12) | 1.5318 | .065 [.061,.070] | .945 | .954 | .045 | 82,152.684 | 82,177.871 |
| | b2. = b1 with random slope | 97.773 (10) | 1.5735 | .027 [.023,.032] | .992 | .992 | .026 | 81,371.435 | 81,405.018 |
| | c1. linear change | 87.629 (8) | 1.6968 | .029 [.024,.035] | .993 | .991 | .026 | 81,370.284 | 81,412.263 |
| | d1. dual change score | 90.422 (9) | 1.5302 | .028 [.023,.033] | .993 | .992 | .026 | 81,357.957 | 81,395.738 |
| | d2. = d1. with free autoproportions | 87.993 (7) | 1.5168 | .031 [.026,.037] | .993 | .989 | .026 | 81,357.057 | 81,403.234 |
| Anxiety (N = 11,866) | a. no growth | 394.299 (14) | 1.782 | .048 [.044,.052] | .962 | .973 | .041 | 92,388.627 | 92,405.442 |
| | b1. constant change | 274.641 (12) | 1.8636 | .043 [.039,.047] | .974 | .978 | .034 | 92,201.793 | 92,227.014 |
| | b2. = b1 with random slope | 84.183 (10) | 1.8006 | .025 [.020,.030] | .993 | .993 | .023 | 91,845.553 | 91,879.181 |
| | c1. linear change | 75.070 (8) | 1.9836 | .027 [.021,.032] | .993 | .992 | .023 | 91,846.877 | 91,888.912 |
| | c2. c1 with random slope | 74.417 (7) | 1.8930 | .028 [.023,.035] | .993 | .990 | .022 | 91,840.838 | 91,887.077 |
| | d1. dual change score | 87.210 (9) | 1.7373 | .027 [.022,.032] | .992 | .991 | .022 | 91,847.481 | 91,885.313 |
| | d2. = d1. with free autoproportions | 59.223 (7) | 1.6371 | .025 [.019,.031] | .995 | .993 | .013 | 91,796.920 | 91,843.159 |
| Depres-sion (N = 11,866) | a. no growth | 734.796 (14) | 2.4894 | .066 [.062,.070] | .889 | .921 | .085 | 103,709.951 | 103,726.765 |
| | b1. constant change | 500.069 (12) | 2.6767 | .059 [.054,.063] | .925 | .937 | .078 | 103,223.249 | 103,248.470 |
| | b2. = b1 with random slope | 148.810 (10) | 2.5869 | .034 [.029,.039] | .979 | .979 | .033 | 102,273.686 | 102,307.314 |
| | c1. linear change | 123.064 (8) | 2.9504 | .035 [.030,.040] | .982 | .978 | .032 | 102,255.814 | 102,297.849 |
| | c2. c1 with random slope | 77.285 (7) | 2.8461 | .029 [.023,.035] | .989 | .985 | .032 | 102,114.687 | 102,160.926 |
| | d1. dual change score | 98.560 (9) | 2.3558 | .029 [.024,.034] | .986 | .985 | .033 | 102,122.918 | 102,160.750 |
| | d2. = d1. with free autoproportions | 37.811 (7) | 1.8828 | .019 [.014,.025] | .995 | .993 | .011 | 101,965.916 | 102,012.155 |
| | d3. = d1. with autoproportion for first component free | 32.006 (8) | 2.2307 | .016 [.010,.022] | .996 | .995 | .012 | 101,964.121 | 102,006.156 |

*(Continued)*

**Table 3.** (Continued)

| | Model | χ2 (df) | SCF | RMSEA [90% CI] | CFI | TLI | SRMR | AIC | sa-BIC |
|---|---|---|---|---|---|---|---|---|---|
| ADHD (N = 11,866) | a. no growth | 507.394 (14) | 1.5164 | .054 [.050,.059] | .968 | .977 | .039 | 82,321.230 | 82,338.045 |
| | b1. constant change | 365.684 (12) | 1.5540 | .050 [.046,.054] | .977 | .981 | .033 | 82,124.070 | 82,149.292 |
| | b2. = b1 with random slope | 61.520 (10) | 1.5685 | .021 [.016,.026] | .997 | .997 | .011 | 81,656.301 | 81,689.930 |
| | c1. linear change | 37.628 (8) | 1.6910 | .018 [.012,.024] | .998 | .998 | .009 | 81,627.436 | 81,669.471 |
| | d1. dual change score | 52.787 (9) | 1.4992 | .020 [.015,.026] | .997 | .997 | .012 | 81,640.943 | 81,678.775 |
| | d2. = d1. with free autoproportions | 39.330 (7) | 1.5232 | .020 [.014,.026] | .998 | .997 | .008 | 81,625.710 | 81,671.949 |
| | d3. = d1. with autoproportion for first component free | 39.970 (8) | 1.5050 | .018 [.013,.024] | .998 | .997 | .008 | 81,623.962 | 81,665.998 |

Note. SR = School Records; SCF = Scaling Factor; RMSEA = Root Mean Square Error of Approximation; CFI = Comparative Fit index; TLI = Tucker-Lewis Index; SRMR = Root Mean Square Residual; AIC = Akaike Information Criterion; sa-BIC = sample size-adjusted Bayesian Information Criterion; The ratio chi/df is > 2 in all models.

the level of the previous time point, was 0.180 ($p$ = .010). The combination of the constant change and proportional change components results in a relatively stable trajectory of school records across the years, with a slight increase (declining performance between time points of 0.015 on average). The intercept was negatively related to the constant change component ($\sigma$ = −0.104, $p$ < .01). The constant change component was not related to gender, but females had better grades at baseline compared to males ($\beta$ = −0.171, $p$ < .01). Fig 1 visualizes the developmental trajectories for males and females separately.

**Anxiety symptoms.** For symptoms of anxiety, the constant change model with random slopes (model b2) appeared to fit the data best. Although the fit criteria were very comparable between the constant change, linear change, and dual change score models (see Table 3), the SB-$\chi^2$ test showed that the constant change model with random slope fitted better compared to the one with fixed slope (SB-$\chi^2$ = 165.35; df = 2; $p$ < .01), which, in turn, fitted better than the no growth model (SB-$\chi^2$ = 147.65; df = 2; $p$ < .01). The linear change model nor the dual change score model fitted better than the constant change model with random slopes (SB-$\chi^2$ = 3.71; df = 2; $p$ = .156 and SB-$\chi^2$ = 0.03; df = 1; $p$ = .864, respectively). In addition, the proportionality components in the dual change score model were not significant ($\beta$ = 0.02; $p$ = .863), which speaks against this model.

The result of the constant change model with random slopes (model b2) showed that the intercept of 0.015 was not significantly different from zero ($p$ = .261) but had a variance of 0.713 ($p$ < .01), showing that students differed in their predicted scores at baseline. The slope was −0.054 ($p$ < .01), reflecting an average decrease in anxiety between time points, which differed between students (residual variance = 0.027; $p$ < .01). The variability in scores at the different time points not accounted for by the model was 0.314 ($p$ < .01). There was a significant covariance between the intercept and slope (−0.048; $p$ < .01). The anxiety scores at baseline did not differ by gender ($p$ = .556), but females showed slightly less decrease in anxiety between time points than males ($\beta$ = 0.040; $p$ < .01).

**Depression symptoms.** The developmental trajectory for symptoms of depression appeared to be reflected best by a dual change score model in which the first proportional change component was freely estimated and the other two were set to be equal (see model d3 in Table 3). This was reflected not only in the fit criteria, but also in the results of the SB-$\chi^2$

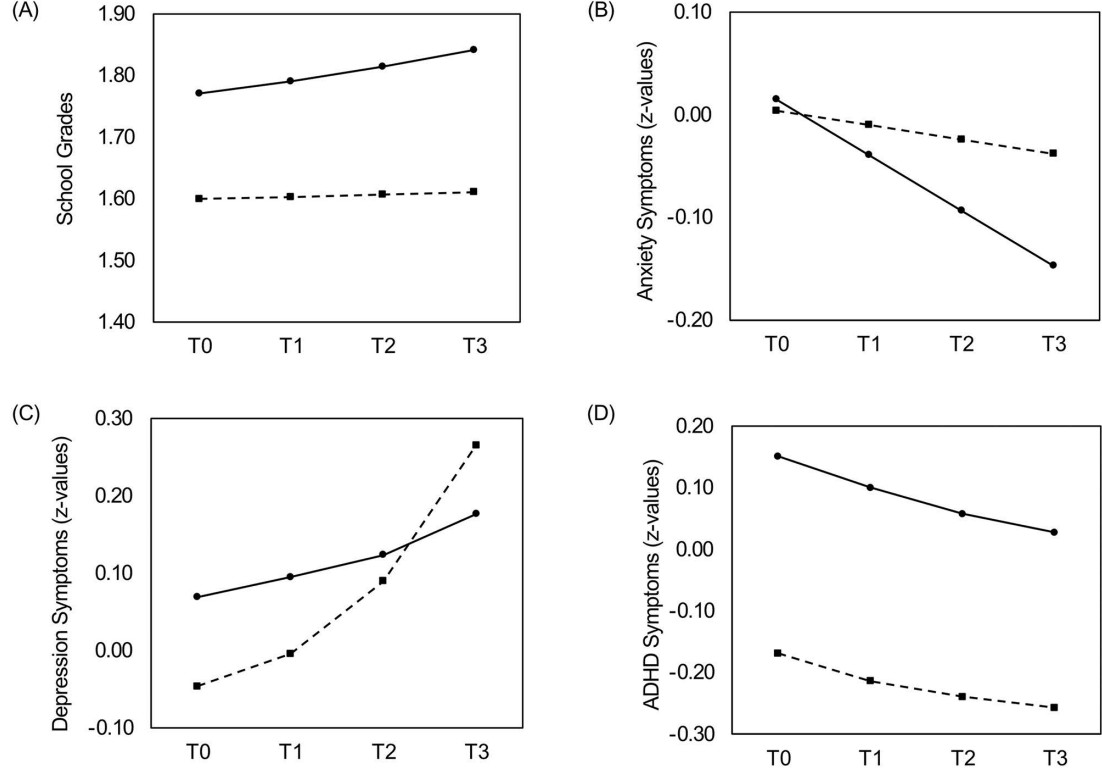

**Fig 1. Model-implied longitudinal trajectories for females (dashed lines) and males (solid lines) across four measurementpoints (T0–T3). (A)** School grades (raw scores); **(B)** Anxiety symptoms (*z*-values); **(C)** Depression symptoms (*z*-values); **(D)** ADHD symptoms (*z*-values).

tests: A dual change score model fitted better than the no change model (SB-$\chi^2$ = 585.01; df = 5; $p$ < .01) and the constant change model with fixed slope (SB-$\chi^2$ = 303.99; df = 3; $p$ < .01) or random slope (SB-$\chi^2$ = 32.74; df = 1; $p$ < .01). Further, the dual change score model fitted better when the proportional change components were freely estimated (SB-$\chi^2$ = 40.14; df = 2; $p$ < .01) and even better when only the first component was freely estimated (SB-$\chi^2$ = 0.04; df = 1; $p$ = .834).

In the resulting model d3, the estimated depression score at baseline was 0.069 ($p$ < .01) with significant between child variability (variance = 0.598; $p$ < .01). The negative constant change component indicated a slight general decrease in depression levels over time (μ = −0.056; $p$ = .011; residual variance = 0.680; $p$ = .008), whereas the positive proportional change components of 1.194 ($p$ < .01; T0 to T1) and 0.882 ($p$ < .01; T1 to T2 and T2 to T3) suggested that individuals with higher symptom levels were at increased risk of symptom maintenance. Overall, this resulted in a progressive increase in symptoms over time. There was a significant negative covariance between the intercept and constant change component (σ = −.631, p < .01). With respect to gender, females showed lower levels of depression at baseline (β = −0.115; $p$ < .01) and a less negative change (β = 0.153; $p$ < .01). The combination of a negative constant change with positive proportional change components, taking into account the gender effects, hints at a progressive increase in symptoms over time for females and only slight increase for males, see Fig 1.

**ADHD.** Although not part of the first research question, the developmental trajectory for ADHD was evaluated as well, as a basis for the trivariate model related to research question 3. The results are comparable to those of depression: The optimal model was a dual change score model in which the first proportional change component was freely estimated and the other two were set to be equal. A dual change score model fitted better than the no change model (SB-$\chi^2$ = 446.10; df = 5; $p$ < .01) and the constant change model with fixed slope (SB-$\chi_2$ = 284.65; df = 3; $p$ < .01) or random slope

(SB-$\chi^2$ = 7.92; df = 1; $p$ < .01). A dual change score model fitted better when the proportional change components were freely estimated (SB-$\chi^2$ = 13,59; df = 2; $p$ < .01), but this freely estimated model did not fit better than one in which only the first component was freely estimated (SB-$\chi^2$ = 0.18; df = 1; $p$ = .672).

In the resulting model d3, the estimated ADHD score at baseline was 0.151 ($p$ < .01) with significant between child variability (variance = 0.764; $p$ < .01). The constant change component was not significant (μ = 0.014; $p$ = .067; residual variance = 0.057; $p$ = .005). The effect of ADHD symptoms at T0 on the subsequent change in symptoms was estimated at −0.243 ($p$ < .01) and the subsequent two proportional change components at −0.284 ($p$ < .01). This resulted in an average trajectory of decreasing symptom levels over time. Higher ADHD scores at baseline were associated with greater increases over time (σ = .142, $p$ < .01). However, when considered together with the negative proportional change components, this effect suggests that children with initially high ADHD symptoms showed less escalation and tend toward stabilization, whereas increases are more pronounced among those starting with low symptom levels. With respect to gender, the results showed that females had clearly lower levels of ADHD at baseline (β = −0.320; $p$ < .01) compared to males and a smaller decrease between time points (β = −0.072; $p$ < .01).

### Relation between school records and anxiety/ depression symptoms

Table 4 shows the model fit for the various bivariate (research question 2) and trivariate (research question 3) latent change score models evaluated.

**School records and anxiety symptoms.** The fit criteria of the no coupling, unidirectional coupling, and full coupling models were not far apart (see Table 4), but the results of the SB-$\chi^2$ tests showed that the no coupling model fitted the data best. More specifically, both unidirectional models (SB-$\chi^2$ = 2.04; df = 1; $p$ = .153; SB-$\chi^2$ = 0.95; df = 1; $p$ = .329) as well as the full coupling model (SB-$\chi^2$ = 3.62; df = 2; $p$ = .164) did not fit better than the no coupling model. The no coupling model appeared to fit best without any alternative model restrictions (setting the slope covariances at zero: SB-$\chi^2$ = 6.01; df = 1; $p$ = .014; slope intercept and covariances across constructs at zero: SB-$\chi^2$ = 4.50; df = 1; $p$ = .034 and SB-$\chi^2$ = 24.67; df = 1; $p$ < .01; all connections over time at zero: SB-$\chi^2$ = 28.99; df = 3; $p$ < .01).

In the no coupling model, the baseline scores for school records and anxiety were strongly related (σ = 0.095; $p$ < .01). Both constant change components were related to each other (σ = 0.003; $p$ = .035), showing that more decrease in anxiety was related to a slightly stronger improvement in grades over the years.

The baseline anxiety score was not significantly associated to the constant change component for school records (σ = −0.015; $p$ = .051). However, children with higher (poorer) grades at baseline showed stronger decrease in anxiety over the years (σ = −0.014; $p$ < .01).

**School records and depression symptoms.** Again, the fit criteria were not far apart (see Table 4), but the results of the SB-$\chi^2$ tests showed that both unidirectional models (SB-$\chi^2$ = 2.28; df = 1; $p$ = .131 and SB-$\chi^2$ = 2.98; df = 1; $p$ = .084) as well as the full coupling model (SB-$\chi^2$ = 5.81; df = 2; $p$ = .055) did not fit better than the no coupling model. Setting the covariance between the intercept of depression and slope for school records at zero improved the model fit (SB-$\chi^2$ = 1.94; df = 1; $p$ = .164). Additionally setting the covariance between both slopes at zero did not improve the model fit (SB-$\chi^2$ = 33.79; df = 1; $p$ < .01), so that model a2 (see Table 4) was chosen as the optimal model.

In model a2, higher depression scores were related to lower achievement at baseline (σ = 0.102; $p$ < .01). The constant change components for depression and grade were positively related (σ = 0.004; $p$ < .01), indicating that children who showed a stronger decrease in depressive symptoms also demonstrated a stronger improvement in grades over time. Children with lower academic performance at baseline showed a stronger decrease in depression over time (σ = −0.105; $p$ < .01).

### The influence of ADHD symptoms

**School records and anxiety symptoms.** Among the trivariate latent change score models for anxiety, school records, and ADHD, the no coupling model showed the best fit. Each of the models for unidirectional or full coupling between

**Table 4. Fit Statistics for the Bivariate and Trivariate Latent Change Score Models.**

| | Model | χ2 (df) | SCF | RMSEA [90% CI] | CFI | TLI | SRMR | AIC | sa-BIC |
|---|---|---|---|---|---|---|---|---|---|
| Anxiety & SR (N = 11,867) | a. no coupling | 199.806 (30) | 1.5057 | .022 [.019,.025] | .993 | .992 | .021 | 172,871.795 | 172,964.275 |
| | b. unidirectional (school records → Δ anxiety) | 197.891 (29) | 1.5043 | .022 [.019,.025] | .993 | .991 | .021 | 172,870.628 | 172,967.311 |
| | c. unidirectional (anxiety → Δ school records) | 198.536 (29) | 1.5085 | .022 [.019,.025] | .993 | .991 | .021 | 172,872.435 | 172,969.119 |
| | d. full coupling | 196.518 (28) | 1.5023 | .023 [.020,.026] | .993 | .991 | .021 | 172,870.176 | 172,971.063 |
| | a1. = a. with slope covariance @0 | 205.947 (31) | 1.4956 | .022 [.019,.025] | .993 | .992 | .021 | 172,876.950 | 172,965.226 |
| | a2. = a. with covariance anxiety intercept - SR slope @0 | 204.043 (31) | 1.5120 | .022 [.019,.025] | .993 | .992 | .021 | 172,877.437 | 172,965.714 |
| | a3. = a. with covariance SR intercept – anxiety slope @0 | 221.085 (31) | 1.4963 | .023 [.020,.026] | .992 | .991 | .023 | 172,899.745 | 172,988.022 |
| | a4. = a. with all connections @0 | 227.973 (33) | 1.4921 | .022 [.020,.025] | .992 | .991 | .023 | 172,905.097 | 172,984.966 |
| Depression & SR (N = 11,867) | a. no coupling | 143.967 (28) | 1.6518 | .019 [.016,.022] | .994 | .993 | .018 | 182,571.309 | 182,672.197 |
| | b. unidirectional (school records → Δ depression) | 141.458 (27) | 1.6565 | .019 [.016,.022] | .994 | .992 | .018 | 182,569.824 | 182,674.914 |
| | c. unidirectional (depression → Δ school records) | 140.480 (27) | 1.6657 | .019 [.016,.022] | .994 | .992 | .018 | 182,569.491 | 182,674.582 |
| | d. full coupling | 137.502 (26) | 1.6693 | .019 [.016,.022] | .994 | .992 | .018 | 182,567.030 | 182,676.324 |
| | a1. = a. with slope covariance @0 | 146.381 (29) | 1.6700 | .018 [.016,.021] | .994 | .993 | .018 | 182,575.952 | 182,672.636 |
| | a2. = a. with covariance depression intercept – SR slope @0 | 145.356 (29) | 1.6620 | .018 [.015,.021] | .994 | .993 | .018 | 182,573.082 | 182,669.766 |
| | a3. = a. with covariance SR intercept – depression slope @0 | 283.123 (29) | 1.5485 | .027 [.024,.030] | .987 | .984 | .025 | 182,769.901 | 182,866.584 |
| | a4. = a. with all connections @0 | 263.706 (31) | 1.6633 | .025 [.022,.028] | .988 | .986 | .025 | 182,766.110 | 182,854.386 |
| | a12. = combination of models a1. and a2. | 169.082 (30) | 1.6422 | .020 [.017,.023] | .993 | .992 | .019 | 182,607.161 | 182,699.641 |
| Anxiety, SR & ADHD (N = 11,867) | a. no coupling | 272.523 (60) | 1.4526 | .017 [.015,.019] | .996 | .994 | .016 | 245,720.890 | 245,897.442 |
| | b1. unidirectional (ADHD → Δ anxiety) | 273.382 (59) | 1.4445 | .017 [.015,.019] | .996 | .994 | .016 | 245,721.915 | 245,902.671 |
| | b2. unidirectional (anxiety → Δ ADHD) | 271.451 (59) | 1.4463 | .017 [.015,.020] | .996 | .994 | .015 | 245,719.609 | 245,900.365 |
| | b3. full coupling anxiety – ADHD | 272.651 (58) | 1.4376 | .018 [.016,.020] | .996 | .994 | .015 | 245,720.985 | 245,905.945 |
| | c1. unidirectional (ADHD → Δ school records) | 272.564 (59) | 1.4490 | .017 [.015,.020] | .996 | .994 | .015 | 245,721.968 | 245,902.724 |
| | c2. unidirectional (school records → Δ ADHD) | 269.105 (59) | 1.4465 | .017 [.015,.019] | .996 | .994 | .016 | 245,716.266 | 245,897.022 |
| | c3. full coupling SR – ADHD | 270.002 (58) | 1.4410 | .018 [.015,.020] | .996 | .994 | .016 | 245,718.092 | 245,903.052 |

*(Continued)*

**Table 4.** (Continued)

|  | Model | χ2 (df) | SCF | RMSEA [90% CI] | CFI | TLI | SRMR | AIC | sa-BIC |
|---|---|---|---|---|---|---|---|---|---|
| Depression, SR & ADHD (*N* = 11,867) | a. no coupling | 258.835 (59) | 1.5508 | .017 [.015,.019] | .995 | .994 | .015 | 254,624.065 | 254,804.821 |
|  | b1. unidirectional (ADHD → Δ depression) | 257.120 (58) | 1.5497 | .017 [.015,.019] | .995 | .994 | .015 | 254,623.125 | 254,808.084 |
|  | b2. unidirectional (depression → Δ ADHD) | 242. 370 (58) | 1.5502 | .016 [.014,.019] | .996 | .994 | .014 | 254,600.390 | 254,785.349 |
|  | b3. full coupling depression − ADHD | 239.264 (57) | 1.5493 | .016 [.014,.019] | .996 | .994 | .014 | 254,597.358 | 254,786.521 |
|  | c1. unidirectional (ADHD → Δ school records) | 258.594 (58) | 1.5443 | .017 [.015,.019] | .995 | .994 | .015 | 254,624.016 | 254,808.976 |
|  | c2. unidirectional (school records → Δ ADHD) | 254.285 (58) | 1.5468 | .017 [.015,.019] | .995 | .994 | .015 | 254,617.984 | 254,802.943 |
|  | c3. full coupling SR − ADHD | 253.032 (57) | 1.5396 | .017 [.015,.019] | .995 | .994 | .015 | 254,616.221 | 254,805.384 |
|  | d. = combination of models b2. and c2. | 241.100 (57) | 1.5466 | .016 [.014,.019] | .996 | .994 | .015 | 254,599.554 | 254,788.718 |

*Note.* SR = School Records; SCF = Scaling Factor; RMSEA = Root Mean Square Error of Approximation; CFI = Comparative Fit index; TLI = Tucker-Lewis Index; SRMR = Root Mean Square Residual; AIC = Akaike Information Criterion; sa-BIC = sample size-adjusted Bayesian Information Criterion.

anxiety and ADHD (SB-$\chi^2$ = 0.50; df = 1; *p* = .479; SB-$\chi^2$ = 1.79; df = 1; *p* = .181; SB-$\chi$2 = 2.07; df = 2; *p* = .356 for models b1 to b3 [see Table 4], respectively) or between school records and ADHD (SB-$\chi^2$ = 0.55; df = 1; ᵖ = .457; SB-$\chi^2$ = 3.64; df = 1; *p* = .056; SB-$\chi^2$ = 3.80; df = 2; *p* = .150 for models c1 to c3 [see Table 4], respectively) did not fit better. This means that ADHD symptoms did not influence subsequent changes in school records or anxiety, or vice versa.

In the resulting trivariate model with no coupling, higher initial ADHD scores as well as more increase in ADHD over time were related to lower grades at baseline (σ = 0.236; *p* < .01 and σ = 0.033; *p* = .019, respectively). More anxiety was related to more ADHD symptoms at baseline (σ = 0.393; *p* < .01). Higher initial ADHD scores appeared related to less decrease in anxiety over time (σ = −0.032; *p* < .01) and increases in ADHD scores over time were related to increases in anxiety symptoms over time (σ = 0.007; *p* < .01). There were no major changes in the model results for the relation between anxiety and school records compared to the model without ADHD.

**School records and depression symptoms.** When focusing on depression symptoms, the trivariate model with school records and ADHD fitted better when including a unidirectional coupling with depression symptoms influencing subsequent changes in ADHD (model b2, see Table 4), compared to the no coupling model (SB-$\chi^2$ = 16.20; df = 1; *p* < .01). Additional coupling between ADHD and subsequent changes in depression (SB-$\chi^2$ = 3.14; df = 1; *p* = .076) or between school records and ADHD (SB-$\chi^2$ = 1.62; df = 1; *p* = .204) did not result in a better fit compared to model b2.

With respect to the relation between ADHD and the depression score at baseline, higher depression scores were related to higher ADHD scores at baseline (σ = 0.339; *p* < .01), but not to changes in ADHD over time (σ = −0.018; *p* = .649). Depression symptoms predicted subsequent changes in ADHD (γ = 0.171; *p* < .01). As mentioned above, an effect from ADHD to school records or depression at the next time point was not found (e.g., model b2 was chosen). There were no major changes in the model results for the relation between depression and school records compared to the model without ADHD.

## Discussion

The aim of the current study was to evaluate the trivariate longitudinal relation between school records, symptoms of anxiety/ depression and of ADHD in children between about 9 and 12 years old. Briefly summarized, the results showed that school records and symptoms of anxiety or depression were related, but school records did not predict these symptoms at the next timepoint, nor the other way around. ADHD was associated with both school records and symptoms of anxiety/ depression. Depression symptoms were a negative leading indicator of the subsequent changes in ADHD.

Research question 1 focused on the longitudinal development of each of the constructs. School records showed a positive linear trend over the four years, alongside negative autoregressive effects of equal magnitude between consecutive time points. In combination, this resulted in a relatively stable, slightly deteriorating trajectory. The hypothesized decline was thus found back, but, contrary to earlier findings [14,15], not only from grade 6 onwards. Anxiety symptoms showed a decrease with age and greater declines in children with higher baseline levels, confirming our hypothesis. For depression symptoms, we found an overall progressive increase in symptoms over time. This increase over time was smaller in children with higher baseline depression levels, hinting at a convergence of trajectories over time. These results confirm our hypothesis only partly. The increase in symptoms over time is as expected, but our results contradict those of earlier research, which found an increase in depression symptoms only starting at age 12, not depending on symptom levels at the previous time point [17] or at baseline [16]. Possibly, this shows that children have started to be vulnerable for depression already in elementary school. However, as the cohorts in the current study were not specifically grade-related, we could not evaluate the effects of school transition. The increase in depression already before the age of 12 could also reflect the effects of the COVID-19 pandemic, which started during the data collection.

Although not the focus of our hypotheses, the current study's results do show gender differences. More specifically, as would be expected, females had lower baseline levels of ADHD. The decrease in symptoms over time was slightly smaller than for males. For depression, females had lower baseline levels, which does not correspond to the frequent finding that females, on average, exhibit more internalizing problems than males [16,56]. However, females showed a stronger increase in symptoms over time than males, resulting in higher symptom levels at the last time point. Anxiety symptoms showed a linear decline over time, which was stronger for males than for females, with no differences in baseline levels. The results of previous longitudinal studies are mixed. For example, McLaughlin and King [16] did not find gender differences in internalizing symptom trajectories. Ohannessian and colleagues [57], however, found a decrease in anxiety disorders in females but not males in middle to late adolescence, which is somewhat comparable to our results. Maybe the relatively young age of the children in our sample (9–12 years) can explain why we did not find higher initial rates for females, as the gender difference emerges only around age 12, at least for depression [58,59], and depression symptoms appear to accelerate earlier in adolescence for females compared to males [60]. However, this does not explain the lower baseline depression levels for females. Clearly, more research into gender differences in internalizing symptom trajectories is needed.

For both anxiety and depression symptoms, the results for research question 2 showed no coupling with school records. In other words, symptoms did not predict school records at the next time point, nor the other way around. This goes against our hypotheses as well as earlier research that showed low achievement to predict higher anxiety levels in the next school year [29,61]. As expected, we did find a negative correlation between symptoms and school records at baseline. Also, the results showed that poorer grades at baseline were related to a stronger decline in anxiety and depression symptoms over the years. Altogether, this means that the level of school records on itself, not its development, is related to anxiety and depression symptoms. This could partly be explained by the way school records develop over time, with improvements overall, but negative effects of one time point to the next. These effects could cancel each other out, resulting in relatively stable school records. Also, although some studies do find a negative relation between anxiety and achievement, some studies actually find a positive relation [e.g., 40], which means that it is likely that other variables play an important role.

Research question 3 focused on the role of ADHD symptoms. For the relation with school records, the results supported our hypotheses in that more ADHD symptoms at baseline were related to poorer grades at baseline. However, we did not find the expected increasing disadvantage caused by ADHD symptoms influencing changes in school records at the next time point. Also as expected, more ADHD symptoms at baseline were related to more internalizing symptoms at baseline as well as a smaller decrease in internalizing symptoms over time. Depression symptoms appeared to predict subsequent ADHD levels, but not the other way around. Contrary to the results of Murray et al. [37], ADHD symptoms did not influence anxiety levels at the next time point, nor the other way around. The study by Murray and colleagues contained a sample of children between 13 and 17 years of age. Possibly, symptoms of both ADHD and anxiety are more stable among younger children and start influencing each other more during adolescence. Another possible explanation is that influences from one time point to the next are dependent on specific ages, as the samples per time point in our study were not as homogeneous in terms of age as those in the study of Murray and colleagues due to the cross-sequential design.

Our hypotheses stated that ADHD symptoms would explain the relation between school records and subsequent anxiety or depression, as well as (for depression) the other way around. This hypothesis could not be evaluated, because these relations were not found in the first place. More in general, there were no clear differences between the models with and without ADHD symptoms in the relations found between school records and anxiety/ depression symptoms. This is not in line with the results of earlier research, which found that depression was more strongly (negatively) related to school records in students with ADHD, compared to those without [6,40]. Also, ADHD symptoms completely accounted for the relation found between reading/ spelling achievement and anxiety symptoms [6].

These contradictory results show the complexity of the underlying phenomena and relations. ADHD symptoms could actually play a role, but it is likely just one of the variables in play. For example, although ADHD symptoms and internalizing problems are clearly related, symptoms of anxiety and/or depression can also arise in the absence of ADHD symptoms and can in that case still be associated with school records. The extent to which school records are affected could depend, for example, on the extent to which the environment of students (teachers, parents) recognizes their symptoms and offer support.

The current study has a number of limitations. First, as mentioned, the ABCD study has a cross-sequential design, as opposed to a cohort-sequential design. The children were 9 years old at the start of the study on average, but the range in both age (8;11–11;1 years) and grade (3–5 for 96% of the sample) was relatively broad. We did not take into account these age and grade differences within cohorts, which means that we could have missed certain age- or grade-specific effects. A cross-sequential design does allow for the evaluation of cohort effects, but because the time range of the baseline measure was broad as well (2016–2018), it was also difficult to take into account the effects of certain events.

This brings us to the second limitation: The influence of the COVID-19 pandemic, at least for part of the sample. The COVID-19 pandemic led to an increase in mental health problems in children and adolescents, especially internalizing symptoms [e.g., 62]. This was also reflected in the sample of the ABCD study, although the increased depression scores were still within the normal range [63]. This will likely have had an influence, especially on the results for research question 1. Due to the complexity of the analyses in terms of the number of variables included, we chose not to take into account the start of the pandemic in our analyses. If it was indeed a confounding variable, this could form part of the explanation why we found earlier increases in depression than expected. It could also explain the different proportional change component for depression from the first to the second time point, as COVID-19 took place at the start of the study. In addition, it might have led to an underestimation of the coupling between depression and school records and/ or ADHD.

Third, we operationalized learning outcomes as the child's school grade in the past year as reported by the parents. Although these parent-reports have been shown to be less valid for children with non-average learning performance, research has supported their use [64]. However, they are not as valid as objective measures of school grades had been. In addition, we used this ordinal variable as a continuous variable in the analyses. This could have resulted in bias in the

form of underestimation of factor loadings, but less so in the estimation of structural relationships between the variables [65]. Also, the use of parent-reported ADHD- and internalizing symptoms is a limitation, because these do not always correspond to self-reported symptoms [66,67] and might underestimate especially internalizing symptoms [68]. The at times non-optimal operationalization of variables is inherent to the use of an existing data set.

Fourth and finally, we used the data from a large study in the USA, where we as researchers do not live ourselves and we were not involved in the data collection. We are thus far removed from the actual object of study and can assess the context factors that might have played a role not as well as a USA resident. However, the study has been logged in a detailed way, so that we were well able to use the data and properly describe the method of study. More in general, the use of the data from the ABCD study has enabled us to evaluate developmental trajectories in a large representative sample, resulting in a high degree of generalizability of the results. The availability of four time points and use of LCSM formed a methodological strength as within person changes are decomposed into different growth components.

Even though we did not find ADHD to explain the relation between school records and internalizing problems, we did find clear associations between school records, internalizing problems, and ADHD symptoms. This is in line with the notion that comorbidity between neurodevelopmental disorders is the rule rather than the exception [69] and with earlier findings on the comorbidity between SLDs and internalizing problems [e.g., 4,5]. More longitudinal studies like the current one are needed to unravel how mental health symptoms and school results develop in relation to each other.

The results of the current study help putting the puzzle together of how neurodevelopmental disorders and their comorbidities can be explained based on the multiple deficit model. This, in turn, can help identifying children with increased risk at an early stage. The fact that we did not find symptoms of depression or anxiety to predict school results at a later time point or vice versa might indicate that they do not influence each other directly. Apart from the finding that depression symptoms predicted subsequent changes in ADHD symptoms, a confounding role of ADHD symptoms in the relation with school records was not found. Therefore, other risk factors, not taken into account in the current study, might be responsible for difficulties in each of the domains. Future research is needed to identify these risk factors. For example, executive function deficits do not only play a role in both SLD and ADHD [70], but are also related to symptoms of depression and anxiety [71]. Although the current study focuses solely on child-related factors, based on the transactional model, child development is the result from a continuous bidirectional effect between the child and its environment [72]. Therefore, risk factors in the environment of the child are essential to take into account as well. For example, parenting characteristics appear to play a role in explaining internalizing problems [73], ADHD symptoms [73] as well as the coping strategies in children with SLD [74].

Altogether, our results showed that school records, internalizing symptoms (anxiety, depression), and ADHD symptoms were all related. However, we did not find direct influences on each other from time point to time point in our longitudinal models, with one exception: Depression symptoms can lead to subsequent increases in ADHD symptoms. These results yield practical recommendations for both clinical and educational practice. In clinical practice, it is essential to be aware of increased risks for ADHD symptoms when a child has depression symptoms. Also, if a child shows increases in ADHD symptoms, this can be a sign of prior depressive symptoms. An advice for teachers is to be aware of an increased risk for internalizing symptoms especially in the case of low grades overall, as opposed to decreasing school results. Also, it is important to recognize signs of depression at an early stage, so that such secondary problems (i.e., ADHD symptoms) can be prevented. Knowledge on such indicators of internalizing problems can be very helpful for increasing the early identification rate. As the name implies, internalizing symptoms are directed inward and therefore relatively hard to recognize for important others, such as parents and teachers. Early identification is, however, essential because internalizing symptoms are related to increased risk for other mental health problems [e.g., 75] and costs for the society [76].

## Supporting information

**S1 Table. Sample Sizes Separately per Time Point, Variable, and by Gender.**
(DOCX)

## Acknowledgments

We would like to thank Prof. Florian Schmiedek from the DIPF for his repeated methodological support, which has been essential for this paper. In addition, we would like to thank the ABCD Research Consortium for sharing their data via the National Institute of Mental Health (NIMH) Data Archive (NDA).

## Author contributions

**Conceptualization:** Linda Visser, Janin Brandenburg.

**Formal analysis:** Linda Visser.

**Validation:** Jan-Henning Ehm, Janin Brandenburg.

**Visualization:** Jan-Henning Ehm.

**Writing – original draft:** Linda Visser.

**Writing – review & editing:** Jan-Henning Ehm, Janin Brandenburg.

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
