## [Decision Letter · Decision Letter 0]

13 Oct 2025

Dear Dr. Janin Brandenburg,

Thank you for submitting your manuscript to PLOS ONE. After careful consideration, we feel that it has merit but does not fully meet PLOS ONE’s publication criteria as it currently stands. Therefore, we invite you to submit a revised version of the manuscript that addresses the points raised during the review process.

Please submit your revised manuscript by Nov 27 2025 11:59PM If you will need more time than this to complete your revisions, please reply to this message or contact the journal office at plosone@plos.org. . . . A rebuttal letter that responds to each point raised by the academic editor and reviewer(s). You should upload this letter as a separate file labeled 'Response to Reviewers'.A marked-up copy of your manuscript that highlights changes made to the original version. You should upload this as a separate file labeled 'Revised Manuscript with Track Changes'.An unmarked version of your revised paper without tracked changes. You should upload this as a separate file labeled 'Manuscript'.

We look forward to receiving your revised manuscript.

Kind regards,

Mu-Hong Chen, M.D., Ph.D.

Academic Editor

PLOS ONE

3. We notice that your supplementary tables are included in the manuscript file. Please remove them and upload them with the file type 'Supporting Information'. Please ensure that each Supporting Information file has a legend listed in the manuscript after the references list.

Reviewers' comments:

Reviewer's Responses to Questions

**Comments to the Author**

1. Is the manuscript technically sound, and do the data support the conclusions?

Reviewer #1: Partly

Reviewer #2: Yes

2. Has the statistical analysis been performed appropriately and rigorously?

Reviewer #1: Yes

Reviewer #2: Yes

3. Have the authors made all data underlying the findings in their manuscript fully available?

Reviewer #1: Yes

Reviewer #2: Yes

4. Is the manuscript presented in an intelligible fashion and written in standard English?

Reviewer #1: Yes

Reviewer #2: Yes

Reviewer #1: To the Authors,

Thank you for the opportunity to review your interesting and methodologically robust study. This note is to explain the reasoning behind my answers to the review questions.

While the manuscript is clearly written and the statistical analyses are appropriate and rigorous (addressing questions 2 & 4), I have indicated that the data only partly support the conclusions at this stage (addressing question 1). This is due to three main reasons:

Measurement Limitations: The use of a single, parent-reported item to measure "learning outcomes" is a significant limitation that impacts the strength of the conclusions drawn about this core construct.

Interpretation of Key Findings: The manuscript presents several significant and compelling gender differences in the results, but these are not interpreted or contextualized in the discussion. This omission leaves a major part of the findings unexplored.

Contextual Factors: The potential confounding impact of the COVID-19 pandemic on the developmental trajectories, particularly for depression, is noted as a limitation but is not sufficiently integrated into the main interpretation of the results.

Addressing these points by expanding the discussion around the limitations of the measurement, interpreting the gender differences, and more deeply considering the pandemic's role will substantially strengthen the manuscript and ensure the conclusions are more firmly supported by the data.

My full, detailed review with specific recommendations for revision is provided in the accompanying attachment. I believe that by addressing these comments, the manuscript will be an excellent contribution to the journal

Reviewer #2: This longitudinal study examined the relationships among adolescents’ learning outcomes, internalizing symptoms (anxiety and depression), and ADHD symptoms using four waves of data from the Adolescent Brain Cognitive Development (ABCD) Study involving 11,867 U.S. children aged 9–12 years. Latent change score modeling showed that academic performance correlated with anxiety and depression but did not predict changes in these symptoms over time. ADHD symptoms were associated with both poorer school records and higher internalizing symptoms; however, they did not explain their interrelation. Depression predicted subsequent increases in ADHD symptoms. The findings highlight the importance of early detection of depressive symptoms. This study was well-conducted and the manuscript is well-written. Some minor comments are listed as below.

1. The study uses four timepoints from the ABCD dataset and latent change score modeling (LCSM), which allows a nuanced examination of within-person changes. This is a methodological strength, particularly for studying developmental trajectories in a large, representative sample.

2. The hypotheses could be better grounded in specific theoretical frameworks (e.g., multiple deficit model or transactional models). The discussion might better integrate these theories to interpret the null findings, particularly why ADHD did not mediate the link between learning outcomes and internalizing symptoms.

3. The exclusive use of parent reports for ADHD, anxiety, and depression may limit validity. Adolescents’ self-reports or teacher evaluations could have provided complementary perspectives, especially for internalizing symptoms, which parents may underreport. This limitation deserves more emphasis in the discussion.

4. While the manuscript concludes with practical recommendations for teachers, these implications are only loosely connected to the actual analytic results. The paper could benefit from a clearer discussion on how observed statistical patterns translate into real-world educational or clinical practice.

.

Reviewer #1: **Yes:** Dr. Chen Hanna RyderDr. Chen Hanna RyderDr. Chen Hanna RyderDr. Chen Hanna Ryder

Reviewer #2: No

---

## [Author Response · Author response to Decision Letter 1]

12 Mar 2026

Response to reviewers

Journal requirements

In response to the request in the decision e-mail, we would like to explain the restrictions to data sharing for this study. We used data from the ABCD study provided by the NIMH. The signed Data Use Certification states:

“Recipients agree to retain control over data and to not distribute, sell, or move data, with or without charge, in any form, to any other individual, entity, or third-party system [...].”

Hence, the data does not belong to us / our institutions and we do not have the right to share the data. We did share all analysis scripts of our study (see https://osf.io/gw8d3/overview?view_only=abcea2aaa99d448a8b7cb068d9f030a0).

Response to reviewer 1

(responses in italics)

Dear Authors,

Thank you for the opportunity to review your manuscript, "The longitudinal relation between adolescents’ learning outcomes and internalizing symptoms: The role of ADHD symptoms". This is an important and timely article that addresses a complex and highly relevant issue. The use of the impressive ABCD Study dataset and the application of sophisticated latent change score models are significant strengths.

Thank you for the extensive review and valuable feedback on our paper.

This note is to explain the reasoning behind my answers to the review questions and to provide detailed feedback for revision. While the manuscript is clearly written and the statistical analyses are appropriate and rigorous, I have indicated that the data only partly support the conclusions at this stage. This assessment is based on several key areas that, I believe, require revision to strengthen the manuscript's overall impact and clarity. My detailed comments are provided below.

Major Points:

1. Operationalization and Measurement of "Learning Outcomes": The sole measure for learning outcomes is a single parent-report item on their child's school grades. While acknowledging the constraints of large-scale datasets, this is a significant limitation that impacts the strength of the conclusions drawn about this core construct.

o Recommendation: Please expand the discussion on this limitation. Specifically, address the potential biases of parent-reported grades versus more objective measures and discuss the implications of treating this ordinal variable as continuous in the models. Furthermore, for consistency, consider changing the keyword "academic achievement" to "parent-reported school grades" or "school records," which are the terms used more consistently throughout the manuscript.

We agree that this operationalization is a limitation and should be discussed accordingly. We have added a paragraph to the discussion (see p. 33-34) and replaced the term “academic achievement” with “school records”(in the keywords as well as the “Data and sample”-section).

2. Interpretation of Key Findings: The manuscript presents several significant and compelling gender differences in the results, but these are not interpreted or contextualized in the discussion. This omission leaves a major part of the findings unexplored.

o Recommendation: The Discussion section should be expanded to include an interpretation of these gender differences. What are the potential theoretical or clinical implications of these findings? For instance, the finding that females show a larger decrease in both anxiety and depression symptoms warrants further exploration in the context of existing literature on gender differences in adolescent mental health.

We did not discuss these gender differences, as they were not the focus of our hypotheses. However, we do agree that this is an interesting part of the results and have therefore added a paragraph to the Discussion section in which we interpret these findings and make a connection with existing literature, see p. 30-31.

3. Contextual Factors: The potential confounding impact of the COVID-19 pandemic on the developmental trajectories, particularly for depression, is noted as a limitation but is not sufficiently integrated into the main interpretation of the results.

o Recommendation: The authors should integrate this potential confounder more deeply into the interpretation of the results in the Discussion. This could help explain why the depression trajectory results contradicted both the authors' hypotheses and previous literature.

Based on this comment, we have noticed that the interpretation of the results with respect to the developmental trajectory for depression was incorrect. More specifically, we concluded that depression symptoms declined, but the descriptive statistics showed an increase. After careful consideration, we have found that the cause for this mistake was a not-optimal coding of gender in the datafile of 1 (male) and 2 (female). We have changed this to 0 and 1, respectively, and were then able to properly interpret the results, which actually showed an increase in depression symptoms over time and thus (partly) confirmed our hypothesis. We have reran all analyses with this adapted coding for gender, which explains the slight changes throughout the Results section. In addition, we have added figures of the developmental trajectories for ease of interpretation (Figure 1). In addition, we incorporated the COVID-pandemic in the interpretation of the results for depression, see p. 30.

4. Clarity of Hypothesis Testing Regarding ADHD's Role: The third research question hypothesized that ADHD symptoms would explain the relation between school records and internalizing symptoms. As the bivariate analyses found no longitudinal coupling between them, this hypothesis could not be directly tested. The current framing might be slightly misleading.

o Recommendation: Consider reframing the third research question and hypotheses in the Introduction. Instead of framing ADHD as an explanatory variable for a presumed relationship, it would be more accurate to state that the study aims to simultaneously examine the trivariate dynamic relationships among the three constructs.

We agree that hypotheses 3.a.iii and 3.b.ii were slightly misleading, because they assumed a certain result for the second research question. We therefore reformulated the aim (“Third, we evaluated the trivariate dynamic relation between ADHD-symptoms, school records and..”). With respect to the hypotheses, we chose to keep the wording such that ADHD-symptoms would explain the relationship, but explicitly added (“if a relation between school records and subsequent anxiety is found, ..”), to take out the misleading character. The reason why we did not want to change the hypotheses is that we prefer not to do so after finishing the study when another solution is available. Also, they connect well to the literature described in the Introduction.

Minor Points:

1. Framing in the Introduction: The introduction begins by discussing Specific Learning Disorder (SLD), but the study focuses on a continuum of symptoms in a general population sample.

o Recommendation: Please make the transition from the discussion of SLD to the study's focus on the continuum of learning outcomes smoother and more explicit.

You are right, this was not completely clear. We have now made the focus on the general population more explicit in the last paragraph of the first part of the introduction.

2. Data from T4: The methods section states that data from T4 were excluded because academic achievement data were missing.

o Recommendation: Please clarify if data for the other key variables (e.g., CBCL) were available at T4 and briefly justify the decision to stop at T3 for all variables rather than using models that can accommodate missing data.

We have added the following clarification: “Although data for the other variables were available, using alternative methods such as imputation was not an option, because school records formed an essential element in the analyses and were missing for all children at T4” (see p. 9).

3. Reporting of Model Fit: In several instances, model selection relied on the chi-squared difference test, even when an alternative model had similar or better fit indices (e.g., RMSEA/CFI).

o Recommendation: It would be helpful to add a brief sentence justifying the reliance on the chi-squared difference test in these specific cases, especially given the large sample size.

We acknowledge that the chi-squared difference test is often significant in the case of a large sample size. However, model comparison was additionally based on the fit indices and on the significance of the proportional change components. We have now better clarified this in the paper on p. 14.

To explain this in a bit more detail, we have summarized the results related to research question 1 in the following table:

Model chi-squared diff. test RMSEA CFI Proportional change components Decision

RQ1 school records d1 b2 c1 / d1 / d2 sign. d1

RQ1 anxiety b2 b2 / d2 d2 n.s. b2

RQ1 depression d3 d3 d3 sign. d3

RQ1 ADHD d3 c1 / d3 c1 / d2 / d3 sign. d3

For school records, the chosen model (d1) does not have the lowest RMSEA, but it does have the highest CFI (just as high as c1 and d2, but d1 is the more parsimonious of these three).

For anxiety, we indeed chose a model that did not have the highest CFI. However, either of the models d would not be a good choice, because the proportional change components were not significant, which speaks for the more parsimonious model b2.

For ADHD, the CFI hints at c1, d2 or d3, but d2 has a lower number of df and thus is less parsimonious. Apart from the chi-squared test and significance of the proportional change components, the AIC- and sa-BIC-values also favor model d3 above c1.

I believe that by addressing these comments, the manuscript will be an excellent contribution to the journal.

We think the adjustments made based on your comments have indeed improved the manuscript.

Response to reviewer 2

(responses in italics)

This longitudinal study examined the relationships among adolescents’ learning outcomes, internalizing symptoms (anxiety and depression), and ADHD symptoms using four waves of data from the Adolescent Brain Cognitive Development (ABCD) Study involving 11,867 U.S. children aged 9–12 years. Latent change score modeling showed that academic performance correlated with anxiety and depression but did not predict changes in these symptoms over time. ADHD symptoms were associated with both poorer school records and higher internalizing symptoms; however, they did not explain their interrelation. Depression predicted subsequent increases in ADHD symptoms. The findings highlight the importance of early detection of depressive symptoms. This study was well-conducted and the manuscript is well-written. Some minor comments are listed as below.

Thank you for the compliments and for your valuable feedback on our paper.

1. The study uses four timepoints from the ABCD dataset and latent change score modeling (LCSM), which allows a nuanced examination of within-person changes. This is a methodological strength, particularly for studying developmental trajectories in a large, representative sample.

Thank you. We had already briefly mentioned this as a strength and have now slightly extended this part, see p. 34.

2. The hypotheses could be better grounded in specific theoretical frameworks (e.g., multiple deficit model or transactional models). The discussion might better integrate these theories to interpret the null findings, particularly why ADHD did not mediate the link between learning outcomes and internalizing symptoms.

We have chosen to keep the introduction as it is in this respect, as we have specified the multiple deficit framework in the beginning and subsequently based the hypotheses on what is known in the empirical literature about the specific relations between the variables of interest. However, we agree that, in the Discussion, we need to come back to this theoretical framework from the Introduction, which was lacking. Therefore, we have added a paragraph at the end of the Discussion in which a link is made to both the multiple deficit model and the transactional model (see p. 34-35).

3. The exclusive use of parent reports for ADHD, anxiety, and depression may limit validity. Adolescents’ self-reports or teacher evaluations could have provided complementary perspectives, especially for internalizing symptoms, which parents may underreport. This limitation deserves more emphasis in the discussion.

We have added the use of parent-report for (both school records and) symptoms of ADHD, anxiety, and depression as a limitation to the Discussion, see p. 33-34.

4. While the manuscript concludes with practical recommendations for teachers, these implications are only loosely connected to the actual analytic results. The paper could benefit from a clearer discussion on how observed statistical patterns translate into real-world educational or clinical practice.

We did try to base the recommendations for teachers on the concrete results of the current study. However, the paragraph (which we now moved to the end of the Discussion) could be better framed, which we did. In addition, we now added recommendations not only for educational, but also for clinical practice, see p. 35-36.

---

## [Decision Letter · Decision Letter 1]

5 Apr 2026

The longitudinal relation between adolescents’ learning outcomes and internalizing symptoms: The role of ADHD symptoms

PONE-D-25-47753R1

Dear Dr. Janin Brandenburg,

We’re pleased to inform you that your manuscript has been judged scientifically suitable for publication and will be formally accepted for publication once it meets all outstanding technical requirements.

Kind regards,

Mu-Hong Chen, M.D., Ph.D.

Academic Editor

PLOS One

Additional Editor Comments (optional):

Reviewers' comments:

Reviewer's Responses to Questions

**Comments to the Author**

Reviewer #1: All comments have been addressed

Reviewer #2: All comments have been addressed

2. Is the manuscript technically sound, and do the data support the conclusions?

Reviewer #1: Yes

Reviewer #2: Yes

3. Has the statistical analysis been performed appropriately and rigorously?

Reviewer #1: Yes

Reviewer #2: Yes

4. Have the authors made all data underlying the findings in their manuscript fully available?

Reviewer #1: Yes

Reviewer #2: Yes

5. Is the manuscript presented in an intelligible fashion and written in standard English?

Reviewer #1: Yes

Reviewer #2: Yes

Reviewer #1: Dear Authors, Thank you for your thorough and thoughtful revision. I am highly impressed by the diligence with which you addressed all of my comments. Your decision to re-examine the data, which led to uncovering the gender coding error, exemplifies excellent and rigorous scientific practice. The revised trajectories now make much more sense, and the expanded discussion regarding measurement limitations, gender differences, and the COVID-19 context has substantially strengthened the manuscript. You have fully resolved all my concerns, and I am very happy to recommend this manuscript for publication. Congratulations on a great piece of work!

Reviewer #2: Thank you for the revision of the article "The longitudinal relation between adolescents’ learning outcomes and internalizing symptoms: The role of ADHD symptoms". My comments have been adequately addressed.

.

Reviewer #1: **Yes:** Dr. Chen Hanna RyderDr. Chen Hanna RyderDr. Chen Hanna RyderDr. Chen Hanna Ryder

Reviewer #2: No

---

## [Editor Report · Acceptance letter]

PONE-D-25-47753R1

PLOS One

Dear Dr. Brandenburg,

I'm pleased to inform you that your manuscript has been deemed suitable for publication in PLOS One. Congratulations! Your manuscript is now being handed over to our production team.

Kind regards,

on behalf of

Dr. Mu-Hong Chen

Academic Editor

PLOS One